# Exercise Affects Mucosa-Associated Microbiota and Colonic Tumor Formation Induced by Azoxymethane in High-Fat-Diet-Induced Obese Mice

**DOI:** 10.3390/microorganisms12050957

**Published:** 2024-05-09

**Authors:** Shogen Yo, Hiroshi Matsumoto, Tingting Gu, Momoyo Sasahira, Motoyasu Oosawa, Osamu Handa, Eiji Umegaki, Akiko Shiotani

**Affiliations:** Department of Gastroenterology, Kawasaki Medical School, Okayama 701-0192, Japan; yoshogen@med.kawasaki-m.ac.jp (S.Y.); gutingting0529@med.kawasaki-m.ac.jp (T.G.); momomo0318@gmail.com (M.S.); o.m.1976-1017@med.kawasaki-m.ac.jp (M.O.); handao@med.kawasaki-m.ac.jp (O.H.); eumegaki@med.kawasaki-m.ac.jp (E.U.); shiotani@med.kawasaki-m.ac.jp (A.S.)

**Keywords:** colon cancer, gut microbiota, mucosa-associated microbiota, exercise, physical activity, *Akkermansia*, *Ruminococcaceae*

## Abstract

The only reliable factor that reduces the risk of colorectal carcinogenesis is physical activity. However, the underlying mechanisms remain unclear. In this study, we examined the effects of physical activity against gut microbiota, including mucosa-associated microbiota (MAM) on azoxymethane-induced colorectal tumors in obese mice. We divided the subjects into four groups: normal diet (ND), high-fat diet (HFD), ND + exercise (Ex), and HFD + Ex groups. The Ex group performed treadmill exercise for 20 weeks. Thereafter, fecal and colonic mucus samples were extracted for microbiota analysis. DNA was collected from feces and colonic mucosa, and V3–V4 amplicon sequencing analysis of the 16SrRNA gene was performed using MiSeq. The HFD group had significantly more colonic polyps than the ND group (ND 6.5 ± 1.3, HFD 11.4 ± 1.5, *p* < 0.001), and the addition of Ex suppressed the number of colonic polyps in ND and HFD groups (ND 6.5 ± 1.3, ND + Ex 2.8 ± 2.5, *p* < 0.05). The HFD group showed significantly lower concentrations of succinic, acetic, butyric, and propionic acids (mg/g) in feces, compared with the ND group (succinic acid HFD 0.59, ND 0.17; acetic acid HFD 0.63, ND 2.41; propionic acid HFD 0.10, ND 0.47; and N-butyric acid HFD 0.31, ND 0.93). In the case of ND, succinic acid and butyric acid tended to decrease with Ex (succinic acid ND 0.17, ND + Ex 0.12; N-butyric acid ND 0.93, ND + Ex 0.74 0.74). Succinic acid, acetic acid, butyric acid, and propionic acid levels in feces were significantly lower in the HFD group than in the ND group; in both feces and mucus samples, *Butyricicoccus* and *Lactobacillus* levels were significantly lower in the HFD group. *Akkermansia* was significantly increased in ND + Ex and HFD + Ex groups. Diet and exercise affected the number of colorectal tumors. Furthermore, diet and exercise alter intestinal MAM, which may be involved in colorectal tumor development.

## 1. Introduction

For the past 30 years, malignant tumors have been the leading cause of death. The incidence and mortality rates of colorectal cancer are increasing. Colorectal cancer (CRC) is the third most commonly diagnosed and deadly cancer worldwide [1]. As one of the most important factors, excess dietary fat intake is intensively associated with increased CRC risk [2]. However, the underlying mechanism between dietary fat intake and CRC development is still largely unclear. Different from other cancer types, CRC directly interacts with trillions of gut microorganisms during tumor development. The composition of gut microbiota is influenced by multiple factors including diet, medication, and genetic alteration, whereas an altered microbial profile can induce dysbiosis and CRC [3]. In particular, the gut microbiota is shown to be perturbed at a very early stage of colorectal tumor formation [4] and becomes aggravated during disease progression [5].

In the development of colorectal adenomas and cancers, the involvement of the intestinal microbiota and its metabolites has recently been reported [6,7,8]. Findings suggest that certain microbial species promote tumorigenesis. Particularly, microorganisms such as *Fusobacterium nucleatum*, *Streptococcus bovis*/*gallolyticus*, *Escherichia coli*, and *Bacteroides fragilis* are abundant in patients with colorectal and adenocarcinomas. These microorganisms may promote the development of colorectal cancer through their ability to adhere to colonic cells, suppress tumor suppressor genes, activate oncogenes, and modulate genotoxicity. In contrast, intestinal microbiota is present not only in the fecal lumen but also in the mucus of the gastrointestinal tract and is referred to as mucosa-associated microbiota (MAM). Interestingly, the microbiota present in the feces and mucus are different [9]. MAM is in contact with the intestinal epithelium and may have a greater influence on colorectal tumors than on the luminal microbiota [5,6,7,8]. As for the MAM of colorectal tumors, *Pseudomonas*, *Helicobacter*, and *Acinetobacter*, which are classified in the genus *Proteobacteria*, have been reported to increase, whereas *Bacteria* have decreased [10].

The WCRF/AICR cites physical activity as the only reliable factor that decreases the risk of colon carcinogenesis [11]. Physical activity is a state that involves skeletal muscle contraction and more energy expenditure than at rest. It can be divided into “activities of daily living” and “exercises”, which are planned and deliberate, aiming to maintain or improve physical fitness. Physical activity prevents colorectal cancer and adenomas, which are precancerous lesions of the colon [12]. According to epidemiological studies and studies using genetic data, increased physical activity may reduce the risk of breast and colorectal cancer [13,14,15,16]. The association between high and low physical activity levels and the risk of digestive cancers (risk ratio (RR) = 0.82, 95% confidence interval (95% CI): 0.79–0.85), colon cancer (RR = 0.81, 95% CI: 0.76–0.87), rectal cancer (RR = 0.88, 95% CI: 0.80–0.98), and colorectal cancer (RR = 0.77, 95% CI: 0.69–0.85) [17]. A large body of epidemiological evidence indicates that people with higher levels of physical activity have a lower risk of developing various cancers than those with lower levels of physical activity. Exercise may reduce the risk of developing colorectal cancer because it prevents obesity and improves gastrointestinal motility. In addition, exercise reduces inflammation and improves immune function, which may reduce the development and progression of cancer. Factors that influence the risk of colorectal cancer include being overweight or obese, physical activity, fiber intake, whole grains, and consumption of red and processed meats. These factors alter the structure and function of the gut microbiota and alter metabolic and immune pathways involved in the development of colorectal cancer. It is also unclear whether changes in the gut microbiota contribute to or are a consequence of the development of colorectal cancer [18].

On the other hand, diet, especially a high-fat diet intake, and obesity have been reported to be associated with colorectal cancer development. In particular, bile acids play an important role in the pathogenesis of colorectal cancer [19]. Intestinal flora also plays an important role in the development and progression of colorectal cancer. High-fat diets affect the metabolism of bile acids, impairing the integrity of the intestinal barrier and affecting intestinal bacteria. High-fat diets stimulate bile acid metabolism and facilitate conversion to tumor-promoting deoxycholic acid by intestinal bacteria [20]. A high-fat diet also alters the composition of intestinal bacteria, causing pathogenic bacteria (such as *Alistipes* sp. *Marseille*-P5997, *Alistipes* sp. 5CPA GH6), causing an increase in the number of pathogenic bacteria and a decrease in beneficial bacteria (e.g., *Parabacteroides distasonis*) [21]. High-fat diets also alter gut bacterial metabolites, causing elevated lysophosphatidic acid, which promotes colon cancer cell growth, and impaired cell adhesion. Furthermore, transplantation of feces from mice fed a high-fat diet into sterile mice induces accelerated colon cell proliferation, impaired intestinal barrier function, and oncogene expression. These studies suggest that a high-fat diet causes intestinal bacterial imbalance and metabolic abnormalities that promote the development of colorectal cancer. However, current research is still limited, and further studies are needed.

This is the first paper to examine the effects of diet and exercise, both of which are considered important in colorectal tumorigenesis, in terms of two intestinal microbiota, fecal and mucosa-associated microbiota (MAM). Studies on the effects of exercise on colorectal tumor suppression from the viewpoint of changes in MAM are nonexistent. We investigated the relationship between exercise-induced changes in MAM and azoxymethane (AOM)-induced colorectal tumors in high-fat diet (HFD)-induced obese mice.

## 2. Materials and Methods

The study was conducted in accordance with the Kawasaki Animal Regulations (Approval No. 22-098). The Balb/c female mice were purchased at 4 weeks, with six to eight animals per group. A 12 h light/dark cycle was maintained, with a room temperature of 22 ± 1 °C, and humidity (55–60%), with ad libitum access to chow and sterilized water under SPF conditions.

Colorectal tumors were used in the AOM-induced model [14,15]. AOM was administered intraperitoneally at 10 mg/kg body weight weekly for a total of six times (6–11 weeks). Colon tumor development was observed every 4 weeks via endoscopy (AVS endoscopy system, OLYMPUS, Tokyo, Japan). Finally, the colon was dissected during autopsy at 28 weeks. We measured colon polyp size using the ruler at 5× magnification.

Mice were divided into four groups according to diet and exercise (Figure 1). The diet was changed after acclimation for 1 week. The diets were a normal diet (ND) and a high-fat diet (HFD 60^Ⓡ^, total calories 5062 kcal/kg; calorie ratio (%): protein 18.2, fat 62.2, and digestible carbohydrates 19.6). The exercise (Ex) consisted of treadmill exercise [16,17]. Running speed was 18 m/min, 30 min/day, and 5 days/week. Exercise was started at 6 weeks of age for 2 weeks and continued for 20 weeks from 8 to 26 weeks.

**As for blood samples,** we measured serum Cholesterol and plasma glucose at 28 weeks. Blood was immediately collected from the heart just before sacrifice, and the separated serum was frozen at −80 °C until analysis. The mice had fasted for nine hours before being sacrificed.

Total RNA was extracted from the colon (tumor area) using an RNase-Free DNase set (QIAGEN, Venlo, NLD, The Netherlands) and reverse-transcribed using the SuperScript^TM^ IV First-Strand Synthesis System (Thermo Fisher Scientific, Waltham, MA, USA) in accordance with the manufacturer’s instructions.

RT-qPCR reactions were performed using a StepOnePlus^TM^ Real-Time PCR Systems (Thermo Fisher Scientific) with PowerUp^TM^ SYBR^TM^ Green Master Mix (Thermo fisher Scientific). The primers used for the RT-qPCR experiments are provided in Table 1. Mouse actin-beta (Actb) expression was evaluated as an internal control. All reactions were performed three times. The PCR conditions were as follows: after initial denaturing at 95 °C for 2 min, 40 cycles at 95 °C for 15 s, and 60 °C for 1 min, followed by a melting-curve analysis (95 °C for 15 s, 60 °C for 1 min, 95 °C for 15 s).

**As for gut microbiota analysis,** fecal and colonic mucus samples were analyzed. Fecal samples were collected at 28 weeks (100 mg feces/mice in 1 mL inhibit EX buffer [approximately 5–6 feces]) from each mouse and stored at −80 °C in 1 mL of EX buffer [Q(IAGEN, Venlo, NLD, The Netherlands]). Colonic mucus was collected from each mouse and stored at −80 °C. For colonic mucus, the colon was incised in the long axis, rinsed lightly with PBS, and the lumen was scraped with a glass slide to collect mucus [18,19]. Bacterial DNA was extracted from the collected samples using the bead-crushing method (QIAamp PowerFecal, QIAGEN), and QIIME was used to perform the V3–V4 amplicon sequencing analysis of 16S rRNA genes at the genus level, identify microorganisms to the genus level, and investigate bacterial composition and diversity. This advanced genomic study was conducted at the Department of Bacteriology, Kyushu University. Taxonomic and functional profiles were further analyzed in STAMP software v2.1.3.

**As to fecal metabolome analysis**, fecal samples were collected at 28 weeks and analyzed for short-chain fatty acids and nonconjugated bile acids (pH-buffered postcolumn conductivity detection method, TechnoSurga).

**Bioinformatics analysis.** Sequence data processing, including chimera check, operational taxonomic unit (OTU) definition, and taxonomy assignment, was performed using QIIME version 1.8.0, USEARCH 6.1, and UCLUST 1.2.22. Open-reference OTU picking was performed at the 97% sequence similarity level against the Greengenes Database, version 13.8. The phyloseq package of R software (version 4.2.2) was used to calculate the observed species, Chao1, and Shannon indices. β-diversity was estimated using the UniFrac metric to calculate the distances between the samples using QIIME version 1.9.1. It was visualized through principal coordinate analysis using R software and statistically analyzed using permutational multivariate analysis of variance using QIIME version 1.9.1.

### Statistical Analysis

Values are presented as mean ± standard deviation or median and 25–75% range, whichever was appropriate depending on whether the data were normally or non-normally distributed. The category data were presented as counts with percentage and analyzed using chi-square test. Continuous variables were tested using the Mann–Whitney U test for comparison between the two groups and by using the Kruskal–Wallis test to compare the 4 groups. Statistical analyses were performed using SPSS (version 25 for Windows, IBM Japan, Ltd., Tokyo, Japan). Statistical significance was set at a *p* value of <0.05.

## 3. Results

### 3.1. Mouse Body Weight Change by Diet and Exercise

HFD increased body weight more than recorded in the ND group (Figure 2). Exercise (Ex) decreased the body weight in both the ND and HFD groups.

### 3.2. AOM-Induced Colorectal Tumor Count

HFD increased the number of colorectal polyps more than ND (ND 6.5 ± 1.3, HFD 11.4 ± 1.5, *p* < 0.001). Exercise suppressed the number of colonic polyps in both the ND and HFD groups (ND 6.5 ± 1.3, ND + Ex 2.8 ± 2.5, *p* < 0.05) (HFD 11.4 ± 1.5, HFD + Ex 5.2 ± 0.8, *p* < 0.01). The polyp suppression effect of exercise was greater with the HFD (Figure 3A,B).

### 3.3. Blood Glucose (BS) and Total Cholesterol Levels

HFD increased the BS and total cholesterol (TC) levels more than ND (BS, ND 105.8; HFD 262.0; *p* < 0.05) (TC, ND 54.0; HFD 106.6; *p* < 0.05) (Figure 4A,B).

### 3.4. Cytokine and Myokine Expression in Colonic Tumors

HFD demonstrated higher IL-6 and TNFα expression in colonic tumors than ND; HFD, ND + Ex, and HFD + Ex showed higher GPR109A expression in colon tumors than ND. ND + Ex revealed predominantly higher SPARC expression in colon tumors than seen in ND (Figure 5A–F).

### 3.5. Short-Chain Fatty Acid (SFA) and Nonconjugated Bile Acids Analyses of Feces

Succinic acid, acetic acid, butyric acid, and propionic acid levels in feces were significantly lower in the HFD group than in the ND group. Succinic acid and butyric acid levels tended to decrease with Ex in the case of ND. In the case of HFD, exercise did not change the amount of the organic acids in the feces. No significant changes were observed in the fecal bile acid concentrations (Figure 6A–E). On the other hand, there were no significant differences in nonconjugated bile acids (Appendix A).

### 3.6. Changes in Fecal Microbiota

The bacterial species that showed significant differences among the four groups were *Akkermansia* and *Alistipes*. In the case of ND (n = 6 per group), those exhibiting significant differences between the two groups with and without exercise were *Akkermansia*, *Candidatus Saccharimonas*, *Eubacterium*, *Staphylococcus*, and *Bacteroides*, which were significantly increased by exercise and significantly decreased by exercise. In the HFD group (n = 6 per group), exercise remarkably increased *Eubacterium* and significantly decreased *Romboutsia* (Figure 7A–E). (Appendix A).

### 3.7. Changes in MAM

*Bacteroides*, *Ligilactobacillus*, and *Clostridia* were remarkably different among the four groups. In the case of ND (n = 6 per group), those that showed significant differences between the two groups in the presence or absence of exercise were *Ruminococcus*, *Akkermansia*, *Staphylococcus*, *Lachospiraceae*, *Oscillospiraceae*, and *Peptococcaceae*, and those that increased significantly with exercise were *Ruminococcus*, *Akkermansia*, *Staphylococcus*, *Lachospiraceae*, *Oscillospiraceae*, and *Peptococcaceae*, whereas a significant decrease was seen in *Romboutsia*. In the case of HFD (n = 6 per group), exercise significantly increased *Muribacuilaceae* and *Akkermansia* and significantly decreased *Lachnospiraceae* and *Oscillospiraceae* (Figure 8A–F) (Appendix A).

## 4. Discussion

In this study, we revealed the changes in MAM following exercise and its inhibitory effect on obesity-induced colorectal tumors in mice. To the best of our knowledge, we demonstrated for the first time that exercise-induced changes in feces and MAM in a colorectal tumor model. Additionally, we confirmed that exercise-induced changes in MAM varied depending on the diet. Interestingly, in both fecal and mucus samples, exercise significantly increased the abundance of *Akkermansia* and *Ruminococcaceae*. Conversely, we also found that some species of bacteria were not common between fecal and mucus samples.

We observed exercise-induced changes in MAM in a mouse colon tumor model for the first time. Exercise remarkably increased *Akkermansia* and *Ruminococcaceae* in mucus samples, while suppressing colon tumorigenesis, which has been reported to impede the thinning of the intestinal mucus layer caused by an HFD [22]. Furthermore, it has been reported that the abundance of *Akkermansia muciniphila* is inversely correlated with inflammatory biomarkers and is associated with a decreased risk of colorectal tumors [23], and that *Akkermansia* and *Lactobacillus* increase during exercise and ND [24,25]. Therefore, *Akkermansia* is expected to be a new probiotic [26] that induces regulatory T cells and suppresses inflammatory cytokines in chronic intestinal inflammation [27,28]. However, there are no reports showing its efficacy in suppressing colorectal tumors, and further studies are required.

Studies on exercise programs for cancer suppression are limited; however, they have been shown to depend on the duration, intensity, and type of exercise with respect to their efficacy [29]. Acute exercise (within 3 weeks) has been shown to be more responsive to NK cells, neutrophils, and macrophages. Conversely, long-term exercise training increases NK cell activity; however, studies regarding other immune cell responses are inconsistent. An 8-week aerobic exercise suppresses FoxP3 + Tre cell accumulation in tumors and increases the CD8+/FoxP3+ ratio within tumors, thereby improving the efficiency of antitumor immunity [30]. Exercise within 3 weeks has also shown (1) an increase in short-chain fatty acids, (2) intestinal epithelial integrity, (3) an increase in mucin-degrading bacteria, and (4) capacity for energy harvesting [31]. It has also been reported that an 8-week aerobic exercise in APC transgenic mouse alters the intestinal microbiota and suppresses colon tumorigenesis [32]. In this study, a unique feature of the exercise was that it was performed for a prolonged period. This exercise model ameliorates metabolic abnormalities caused by an HFD. The present study involved long-term aerobic exercise of some intensity for 20 weeks. Similar exercises improve metabolism [33,34].

Regarding exercise-induced changes in intestinal short-chain fatty acids, butyrate has been reported to have anti-inflammatory and antitumor effects [35], and treadmill exercise in mice has been shown to increase the concentration of short-chain fatty acids in the colon [36]. Also, exercise increased n-butyrate concentrations in the rat cecum [23,37,38,39]. In human studies, exercise increased fecal acetate, propionate, and butyrate in lean individuals but not in obese ones [24,40]. Thus, exercise-induced short-chain fatty acid concentrations are altered by diet and body size. In this study, we demonstrated that the effects of exercise are altered by diet. However, in this study, the short-chain fatty acid analysis differed from the fecal analysis. One reason for this could be that the fecal material in the cecum and secreted outside the body may be different. We should analyze the different compositions between feces and luminal feces in the future.

In this study, we found that different diets lead to different exercise-induced changes in the gut microbiota. Diet is more important than exercise in affecting gut bacteria [16,39]. HFDs are known to increase colorectal tumors. These effects have been reported to be due to the composition and metabolism of the intestinal microflora associated with bile acids [40,41]. However, in this study, we observed a colorectal tumor-suppressive effect of exercise regardless of diet. This may be due to the effect of exercise over a longer period of time than described in previously reported exercise interventions or to different durations. Because changes in intestinal bacteria due to exercise depend on the diet, examining the diet and the effect of concomitant therapy to maximize the effect of exercise is necessary.

The characteristics of colonic MAM in colorectal tumors in HFD-induced obese mice were investigated. The HFD group had significantly more colonic polyps than the ND group. Some studies have suggested a role for gut mucosa-associated microbiota in the development of obesity, but the mechanisms involved are poorly defined. Quinyun Mao, et al. investigated the differences between the diversity of luminal and mucosa-associated microbial communities in obese mice [41]. The study found differences in microbial composition at the phylum and genus level between the microbial flora from colonic contents and that in colonic mucosa, although they had similar richness, evenness, and overall structure. At the phylum level, the colonic contents showed a higher abundance of *Bacteroidetes*, while colonic mucosa had a higher abundance of *Firmicutes* and *Proteobacteria*. At the genus level, the butyrate-producing bacteria *Lactobacillus* were more abundant in colonic contents, while the Gram-negative genera *Helicobacter*, *Sphingomonas*, and *Desulfovibrio* were relatively abundant in the colonic mucosa. Furthermore, Xu, et al. investigated the impact of the gut mucosa-associated microbiota on obesity, and related metabolic disorders have been investigated in a porcine model of metabolic syndrome [42]. Association analysis revealed that certain bacteria, such as *Lactobacillus johnsonii* in the duodenum; *Actinobacillus indolicus* in the jejunum; *Acinetobacter johnsonii* in the ileum; *Clostridium butyricum* in the cecum; *Haemophilus parasuis* in colon; and bacterium *NLAEzlP808*, *Halomonas taeheungii*, and *Shewanella* sp. *JNUH029* in the rectum, play crucial roles in adiposity. This study found that the colonic feces of the HFD group had higher abundance of *Romboutsia*, *Lactococcus*, and *Faecallibaculum*, as well as *Lachnospiraceae*. In contrast, the colonic mucosa had higher abundance of *Osillospiraceae* and *Lachnospiraceae*. Additionally, exercise in the HFD group led to a significant increase in *Muribacuilaceae* and *Akkermansia* and significantly decreased *Lachnospiraceae* and *Oscillospiraceae.* This report describes the changes in the bacterial flora in feces and mucus due to exercise under HFD. However, further studies are needed to confirm these findings.

The study found no evidence of changes in bile acid metabolism being associated with colorectal polyps in this experimental model. However, previous studies have reported a link between high-fat diets and bile acid metabolism relative to the risk of colorectal tumors. The previous research reported that an HFD may increase the risk of developing colorectal precancerous lesions and adenomatous polyps and exacerbate colorectal tumor progression [43]. According to Li Liu et al., deacidified bile acids (DCA) can cause low-grade inflammation in the intestine, disrupting the physical and functional barriers of the mucosa and worsening the spasticity of intestinal tumors [19].

This study has some limitations. First, the sample size was small. Microbiome studies have great variability, and a higher sample size is preferred. Secondarily, we did not measure the amounts of the products of our target genes in the serum. Thus, we did not see the differences between tissue and serum in this study. Additionally, although we observed the changes in the fecal microbiota and the MAM, it is not clear whether they were the cause or the effect. Further studies are needed to confirm these findings.

## 5. Conclusions

In this study, we investigated the effects of physical activity against gut microbiota, including MAM, on AOM-induced colorectal tumors in mice. Diet and exercise affected the number of colorectal tumors as well as the fecal microbiota and MAM. The HFD group had significantly more colonic polyps than did the ND group, and the addition of Ex suppressed the number of colonic polyps in the ND and HFD groups. In both feces and mucus samples, *Butyricicoccus* and *Lactobacillus* levels were significantly lower in the HFD group. *Akkermansia* was significantly increased in the ND + Ex and HFD + Ex groups. Furthermore, diet and exercise alter intestinal MAM, which may be involved in colorectal tumor development.

## Figures and Tables

**Figure 1 microorganisms-12-00957-f001:**
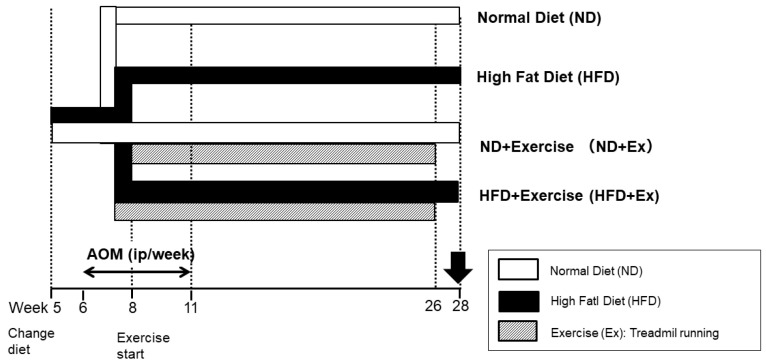
Study design. Four groups: Normal diet (ND), High-fat diet (HFD), ND + Exercise (Ex), and HFD + Ex. AOM: azoxymethane, ip: intraperitorial injection.

**Figure 2 microorganisms-12-00957-f002:**
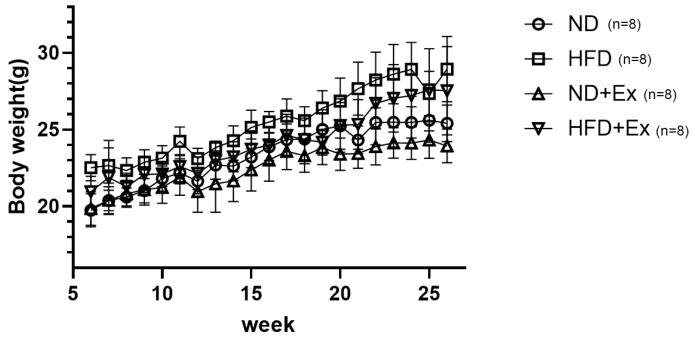
Mouse body weight change. At 26 weeks, the highest weight was in the high-fat diet (HFD) group and the lowest in the normal diet plus exercise (ND + Ex) group.

**Figure 3 microorganisms-12-00957-f003:**
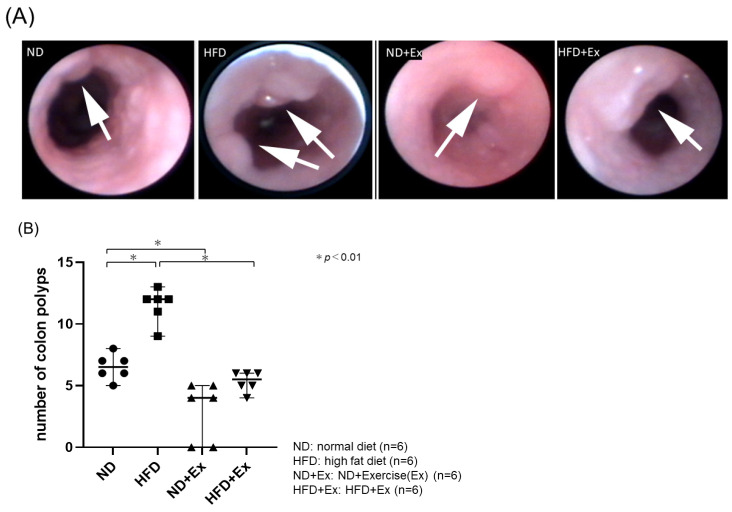
(**A**) Endoscopic findings; (**B**) comparison of number of total colonic polyps; and (**C**) comparison of number of colonic polyps and polyp size distribution. The HFD group had the highest number of polyps and the largest polyps. The ND + Ex group had the fewest polyps and the smallest polyp size.

**Figure 4 microorganisms-12-00957-f004:**
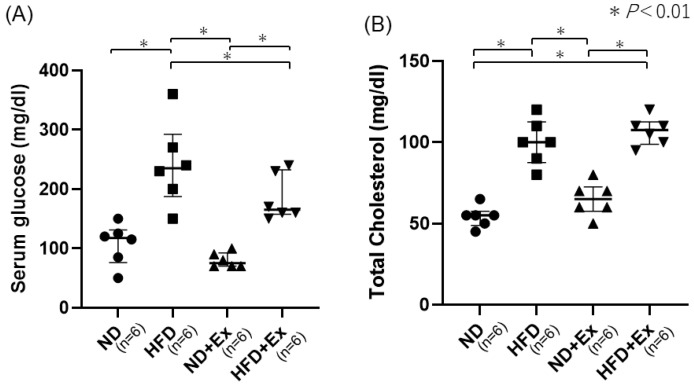
(**A**) Blood glucose and (**B**) total cholesterol levels. Serum blood glucose levels were higher in the HFD group than in the ND and decreased with the addition of Ex. The decrease associated with Ex was particularly marked for the HFD.

**Figure 5 microorganisms-12-00957-f005:**
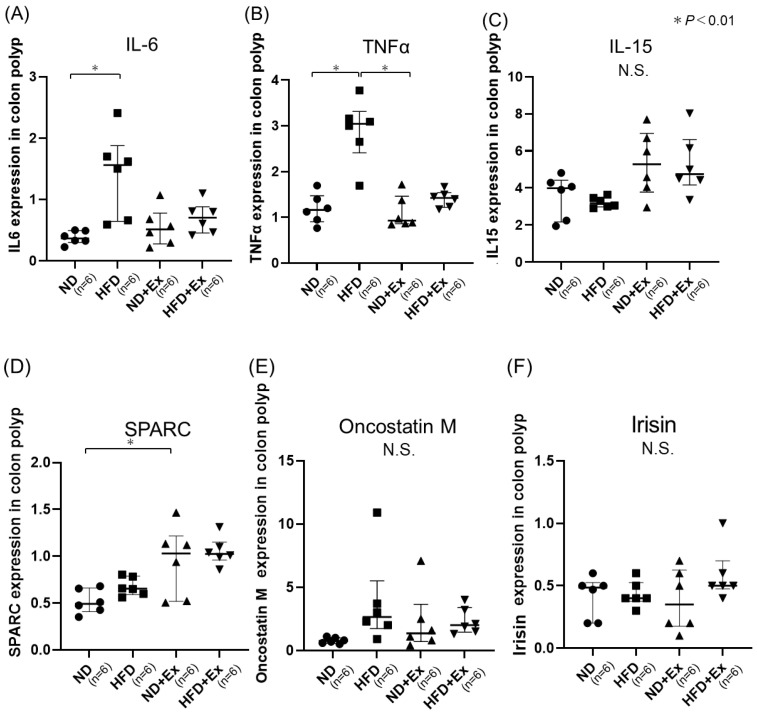
Expression of cytokines and myokines by qPCR in colonic tumors. (**A**) IL-6; (**B**) TNF-alpha; (**C**) IL-15; (**D**) SPARC; (**E**) Oncostatin M; and (**F**) Irisin. The HFD group had significantly lower TNFα and IL-6 in colon tissue than the ND group. Exercise did not affect cytokines, and SPARC was significantly higher only in the ND group. ● ND group, ■ HFD group, ▲ ND + Ex group, ▼ HFD + HFD + Ex group.

**Figure 6 microorganisms-12-00957-f006:**
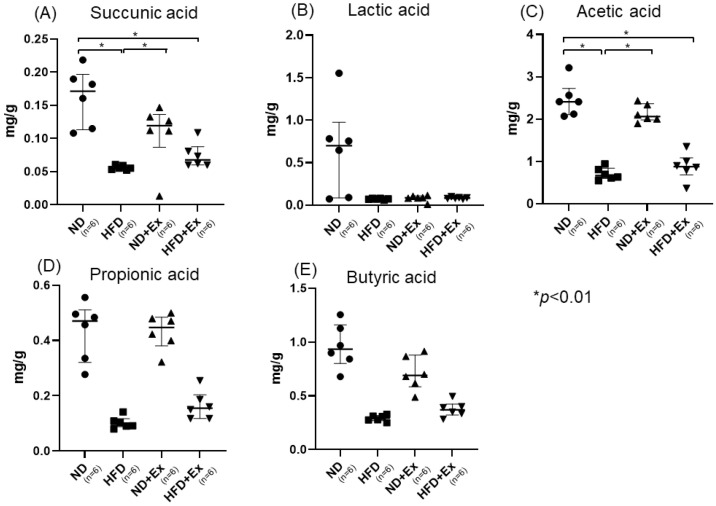
Fecal SFA levels: (**A**) succinic acid; (**B**) lactic acid; (**C**) acetic acid; (**D**) propoonic acid; and (**E**) Butyric acid. Succinic acid and acetic acid were significantly lower in the HFD than in the ND. Succinic acid and acetic acid were significantly increased in the ND+Ex, as compared to HFD.

**Figure 7 microorganisms-12-00957-f007:**
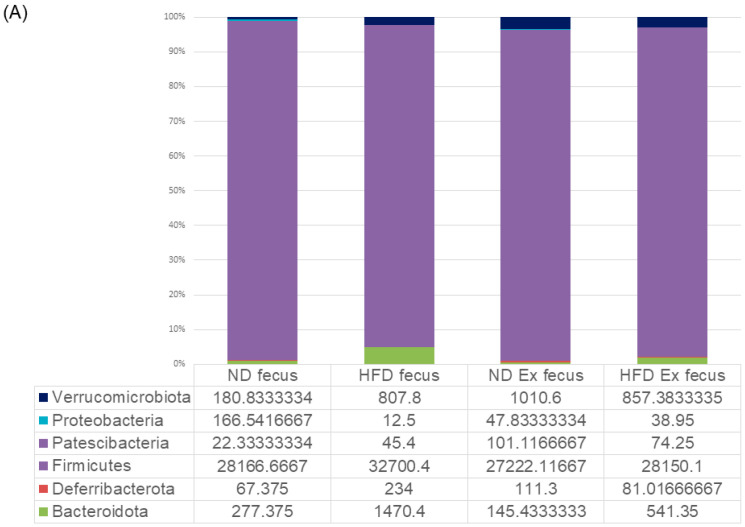
Fecal microbiota analysis: (**A**) taxonomic analysis; (**B**,**C**) comparison of 4 groups (list of bacteria species that showed significant differences among the four groups); (**D**) bacteria species that showed significant differences between the two groups of ND and ND + Ex; and (**E**) bacteria species that showed significant differences between the two groups of HFD and HFD + Ex.

**Figure 8 microorganisms-12-00957-f008:**
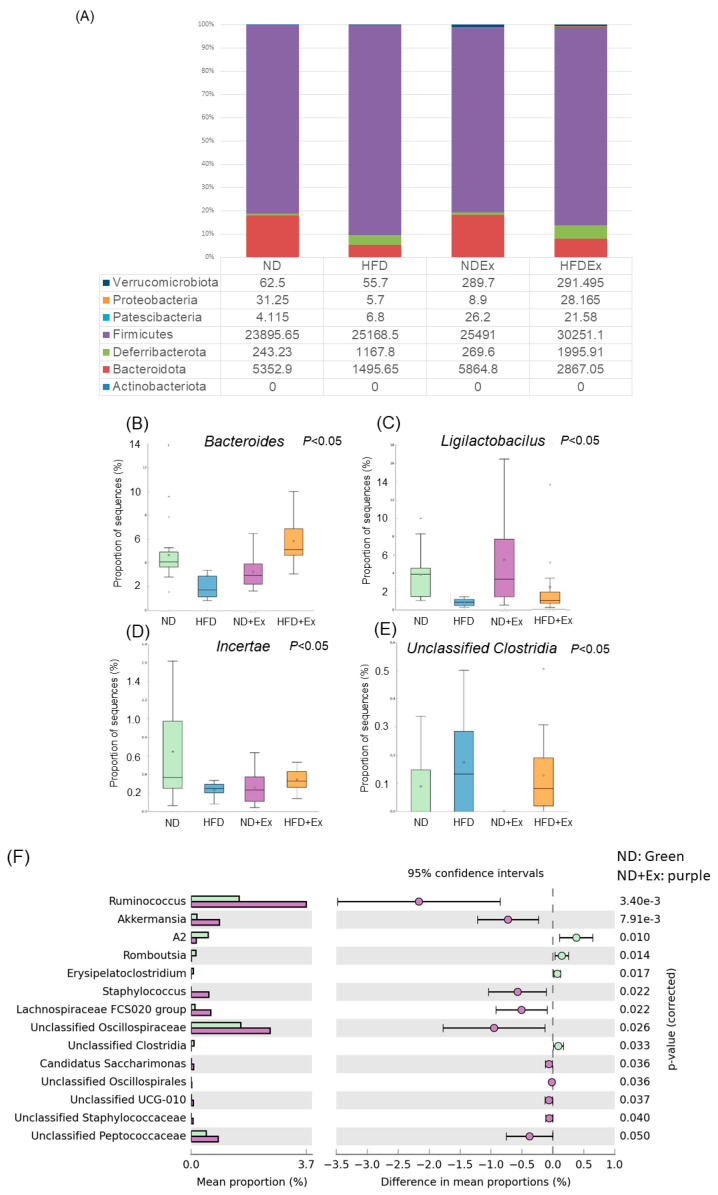
Mucosa-associated microbiota (MAM) analysis: (**A**) taxonomic analysis; (**B**–**E**) comparison of 4 groups (list of bacteria species that showed significant differences among the four groups); (**F**) bacteria species that showed significant differences between the two groups of ND and ND + Ex; and (**G**) bacteria species that showed significant differences between the two groups of HFD and HFD + Ex.

**Table 1 microorganisms-12-00957-t001:** PCR primer sequences.

Target Gene		Sequence	Length
**IL-6**	FP	CGGCCTTCCCTACTTCACAAGTCCG	66
	RP	CAGGTCTGTTGGGAGTGGTATCC	
**TNF-alpha**	FP	CCACCATCAAGGACTCAAATGG	74
	RP	CCTTTGCAGAACTCAGGAATGGACATTCG	
**IL-15**	FP	CATCCATCTCGTGCTACTTGTG	112
	RP	GCCTCTGTTTTAGGGAGACCT	
**SPARC**	FP	CCACACGTTTCTTTGAGACC	95
	RP	GATGTCCTGCTCCTTGATGC	
**Oncostain M**	FP	GTGGCTGCTCCAACTCTTCC	81
	RP	AGAGTGATTCTGTGTTCCCCGT	
**Irisin**	FP	GAGCCCAATAACAACAAGG	242
	RP	GAGGATAATAAGCCCGATG	
**Actb**	FP	CACTGTCGAGTCGCGTCC	102
	RP	CGCAGCGATATCGTCATCCA	

## Data Availability

The raw data supporting the conclusions of this article will be made available by the authors on request.

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
