# Peer review of "Exercise Affects Mucosa-Associated Microbiota and Colonic Tumor Formation Induced by Azoxymethane in High-Fat-Diet-Induced Obese Mice"

_microorganisms, 2024, doi:10.3390/microorganisms12050957_

Round 1
Reviewer 1 Report
Comments and Suggestions for Authors
Dear Authors,
Thank you for providing this manuscript. Please find my comments per line and questions below:
Line 70 and 71: As colorectal cancer is doubled here, could you please give the difference between this two leading to the different values?
Line 91: please set the bacteria names to italic and place a space between Alistipes and sp..
Study design: Did you control the calorie intake of the mice?
Could you please give some parameters about the housing of your animals? Where they ad libitum fed? Did you use a 12 hours day/night rhythm? Did they live under SPF conditions? These parameters influence the health and the behaviour of the animals and with this the feeding and so the microbiome.
Did you measure the amount of the products of your target genes in serum? Could you see differences there?
Line 165: As your dataset contains only 6-8 mice per group, I would suggest using only non-parametric tests.
When/where did you draw the blood samples and how?
Line 168: Could you please specify, which two groups you mean here using the Mann-Whitney-U test as you described 4 groups above. The Mann-Whitney-U test is not possible for comparing 4 groups. Please clarify.
Figure 2: Could you please use symbols, which make it easier to distinguish between the groups? It is very hard to see. Maybe you could use only frames instead of filled symbols or you could draw the figure a little bit bigger as everything is so close together?
Figure 3: Could you please add a graph with the polyp size? How was the size measured?
Line 189: Could you please give some more details about the BS measurement as the BS is directly influenced by food intake, stress and so on? Where the animals fasted? How long?
Results: Could you please give the exact n per group you evaluated? Materials and methods only says 6-8 per group but looking at the graphs there is a maximum of 6 dots, sometimes even less. So please place the n per group for every graph in the figure legend as the number per dots differ per group in different graphs.
Figure 6A: One significance is missing on the bracket, please ad.
Line 220: Saccahrimonas seems to be a typo. Please correct. Candidatus is a status and not a genus. It was introduced to bacterial taxonomy to accommodate uncultured taxa defined by analyses of DNA sequences. Please add some more of the name to allow the reader to follow. As written in line 222 Candidatus Saccharimonas could be ment which would reduce the groups to 3. Please check.
Figure 7: This figure seems to be mixed up: B,C,D: The figure legend does not fit the figure. Please clarify. The figure legend for E and F is missing. Please add. Please also check Figure 8 concerning the same points.
Line 257 ff and line 262ff contain the same results. Please rephrase.
Line 268 and line 270 also contain the same finding. Please rephrase.
Line 304: Please show your data for the tumor-suppressive effect of exercise regardless of diet. Did you check tumor size, growth rate and so on?
316/317/318 and following: please set the bacteria names to italic
Line 341 and following: As you did not measure these parameters, this paragraph is not needed here.
Line 349 ff: I would suggest not to offer your plans to others. Maybe they will take your ideas and get the results before you can publish them.
kind regards
Comments on the Quality of English LanguageDear Authors,
The manuscript is written in propper English. Sometimes information is doubled. In some cases a few more information is needed. Please see my comments above.
kind regards
Author Response
Thank you very much for the reviewer’s advice and comments. According to reviewer's comments, we rewrote and added new sentences, and new parts that were in bold. We believe that the manuscript has been significantly improved by your comments. We hope the revised manuscript is suitable for publication in Microorganism.

Reviewer 2 Report
Comments and Suggestions for Authors
The study investigates the impact of exercise on mucosa associated microbiota in obesity-induced colorectal cancer in mice. The study is of great relevance on the field, although results should be interpreted carefully.
- Figure 3A. Mark the the polyps in the images.
- Figure 7 and 8. Letters should be bigger, it is difficult to read.
Author should be clear in discussion and conclusions about the limitations of the results, specially about the sample size. Microbiota studies have great variability and higher sample size is prefered.
Conclusions should address more of the relevant findings of the manuscript based on the results. It is repeating the last paragraph of discussion.
The planned fecal transplant study is promising but should include a comprehensive analysis of the transplant recipients' microbiota and tumor development to confirm the inhibitory effect of exercised donor microbiota on colorectal tumors.
Comments on the Quality of English LanguageNo comments
Author Response

(The authors gave the same response as above.)

Round 2
Reviewer 1 Report
Comments and Suggestions for Authors
Dear Authors,
Thank you for taking my suggestions into account and providing this improved manuscript. I would like to ask you for some small points to be found in the by line comments below:
Line 29/387: Please clarify if you use family or genus here. At the moment it is a mix of both.
Line 32: I would suggest not to offer your plans to others. Maybe they will take your ideas and get the results before you can publish them.
Line 121: please delete “. The animals were allowed free” to connect these sentences.
Line 127: Could you please give the magnification where the colon polyp size was measured?
Figure 5: Please stay consistent with the writing of the headlines/axes. Here you switch from capital to lowercase letters.
Figure 7 B/C, 8 B/C/D/E: I am very sorry but I cannot read the y axis label and compartments (Figure 7) or x and y axis for Figure 8. Please also enlarge the p values in figures 8 B/C/D/E.
Paragraph 3.6/3.7: Please give the number of animals you analysed. For example (line 261): “In the case of ND (n=6 per group),…” ; (Line 266) “In the case of HFD (n=6 per group),…”
Line 356/357: please set NLAEzlP808 and sp. JNUH029 to italic as they belong to bacterial names
Line 369: Did you check for colorectal precancerous lesions in histological sections?
Line 374: please correct to study limitations
Line 375 please correct to studies
Line 375: grate seems to be a typo; great? Please check.
Data availability statement: Please check, it is missing.
Kind regards
Comments on the Quality of English LanguageDear Authors,
Please find my comments above.
kind regards
Author Response
Thank you very much for the reviewers advice and comments. According to reviewer's comments, we rewrote and added new sentences, and new parts are in bold. We believe that the manuscript has been significantly improved by your comment. We hope the revised manuscript is suitable for publication in Microorganism.
